# The Six-Item Version of the Internet Addiction Test: Its Development, Psychometric Properties, and Measurement Invariance among Women with Eating Disorders and Healthy School and University Students

**DOI:** 10.3390/ijerph182312341

**Published:** 2021-11-24

**Authors:** Amira Mohammed Ali, Amin Omar Hendawy, Abdulaziz Mofdy Almarwani, Naif Alzahrani, Nashwa Ibrahim, Abdulmajeed A. Alkhamees, Hiroshi Kunugi

**Affiliations:** 1Department of Behavioral Medicine, National Institute of Mental Health, National Center of Neurology and Psychiatry, 4-1-1, Ogawahigashi, Kodaira, Tokyo 187-8553, Japan; 2Department of Psychiatric Nursing and Mental Health, Faculty of Nursing, Alexandria University, Smouha, Alexandria 21527, Egypt; 3Department of Biological Production, Tokyo University of Agriculture and Technology, Tokyo 183-8509, Japan; amin.hendawy@gmail.com; 4Department of Animal and Poultry Production, Faculty of Agriculture, Damanhour University, Damanhour 22516, Egypt; 5Department of Psychiatric Nursing, College of Nursing, Taibah University, Janadah Bin Umayyah Road, Tayba, Medina 42353, Saudi Arabia; ammrwani@taibahu.edu.sa; 6Department of Medical Surgical Nursing, College of Nursing, Taibah University, Janadah Bin Umayyah Road, Tayba, Medina 42353, Saudi Arabia; Nzahrani@taibahu.edu.sa; 7Department of Psychiatric and Mental Health Nursing, Faculty of Nursing, Mansoura University, Mansoura 35516, Egypt; Nashwa_2005@mans.edu.eg; 8Department of Medicine, College of Medicine and Medical Sciences, Qassim University, Al Qassim, Buraydah 52571, Saudi Arabia; A.alkhamees@qu.edu.sa; 9Department of Psychiatry, School of Medicine, Teikyo University, 2-11-1 Kaga, Itabashi-ku, Tokyo 173-8605, Japan; hkunugi@med.teikyo-u.ac.jp; 10Department of Mental Disorder Research, National Institute of Neuroscience, National Center of Neurology and Psychiatry, 4-1-1, Ogawahigashi, Kodaira, Tokyo 187-8502, Japan

**Keywords:** coronavirus disease 2019, anorexia nervosa/binge eating/eating disorder, women, school children, university students, factorial structure/psychometric properties/structural validity/validation, Internet Addiction Test/six-item Internet Addiction Test, invariance, internet dependence/problematic internet use

## Abstract

Internet addiction (IA) is widespread, comorbid with other conditions, and commonly undetected, which may impede recovery. The Internet Addiction Test (IAT) is widely used to evaluate IA among healthy respondents, with less agreement on its dimensional structure. This study investigated the factor structure, invariance, predictive validity, criterion validity, and reliability of the IAT among Spanish women with eating disorders (EDs, *N* = 123), Chinese school children (*N* = 1072), and Malay/Chinese university students (*N* = 1119). In school children, four factors with eigen values > 1 explained 50.2% of the variance, with several items cross-loading on more than two factors and three items failing to load on any factor. Among 19 tested models, CFA revealed excellent fit of a unidimensional six-item IAT among ED women and university students (χ^2^(7) = 8.695, 35.038; *p* = 0.275, 0.001; CFI = 0.998, 981; TLI = 0.996, 0.960; RMSEA = 0.045, 0.060; SRMR = 0.0096, 0.0241). It was perfectly invariant across genders, academic grades, majors, internet use activities, nationalities (Malay vs. Chinese), and Malay/Chinese female university students vs. Spanish women with anorexia nervosa, albeit it was variant at the scalar level in tests involving other EDs, signifying increased tendency for IA in pathological overeating. The six-item IAT correlated with the effects of internet use on academic performance at a greater level than the original IAT (*r* = −0.106, *p* < 0.01 vs. *r* = −0.78, *p* < 0.05), indicating superior criterion validity. The six-item IAT is a robust and brief measure of IA in healthy and diseased individuals from different cultures.

## 1. Introduction

The use of the internet for socialization and gaming has dramatically increased among children, adolescents, and young adults during the last two decades as a result of the expansion of internet technology [1,2]. Excessive internet use, particularly among individuals with special emotional needs, has brought about several negative consequences such as decreased sleep duration, cyberbullying, nomophobia, and internet addiction (IA) [3,4,5]. IA is a maladaptive pattern of excessive or problematic use of the internet for nonessential, personal internet activities (e.g., gaming, social networking, gambling, and online sex) that increase the time spent online and cause remarkable alterations in one’s life [6,7].

The current coronavirus disease 2019 (COVID-19) pandemic has been associated with an increase in the prevalence and severity of IA among the general population, especially youth and individuals with poor social support [4,8]. This is because the pandemic and lockdown restrictions have increased anxiety and distress levels in the general population who tend to problematically use certain online applications as a way to compensate for negative emotions [9]. Evidence documents a considerable increase in the duration of internet in the general population secondary to anxiety associated with fear of contracting infection as well as social isolation, a key protective measure against COVID-19 [4,10]. Nation-wide surveys in countries intensively struck by COVID-19 such as China emphasize increased frequency and duration of internet usage among children and adolescent during the pandemic, especially for recreational activities. Age, gender, depression, and stress are key predictors of IA [11]. Among Italian students, fear of COVID-19 was common; it positively correlated with IA, depression and anxiety. In age- and gender-adjusted analysis, fear of COVID-19 mediated the relationship between anxiety and IA [10]. In a large-scale study in Germany, 71.4% of the participants reported increased use of online media during the lockdown [9]. Internet use activities varied remarkably across genders: males reported increased gaming activities and online sex while females reported increased engagement in social networking, video streaming, and information research [9]. Investigations of daily internet habits and social media use among adolescents and young adults from India, Mexico, the Philippines, and Turkey during COVID-19 show that psychological distress, self-esteem, loneliness, and escapism are consistent predictors of IA. Evident cultural differences were noted by considerable variations in IAT scores across countries. Internet use activities also varied, with higher use of social media in Philippines and increased gaming activities in Turkey [12]. It is not clear whether increased IA during COVID-19 is a functional and time-limited phenomenon or it is a trend toward elevated occurrence of IA [9].

Griffiths’ addiction components model proposes that both drug addiction and addictive behaviors, including IA, comprise a set of six distinct common components: salience, mood modification, tolerance, withdrawal, conflict, and relapse [13]. In fact, a parsimonious IA components model based on Griffiths’ model has been reported to organize the self-reported behavioral components of IA [14]. In parallel, the DSM-5 details the diagnostic criteria displayed by individuals with problematic internet use: salience indicated by preoccupation, repeated and uncontrolled use of the internet despite its negative psychosocial effects expressed as conflicts in work and academic relationships, loss of interest in other recreational activities, using the internet to escape negative emotions or to improve mood, misleading others regarding the amount of time spent online, intolerance—a need for increased internet use to achieve the previous desired effect, withdrawal reactions (e.g., anxiety and depression) following deprivation of the internet, and relapse—tendency to revert to earlier use following abstinence [3,6,15].

IA is associated with poor academic/work performance, dysphoric mood, dysfunctional sleep, loneliness, and deficient real world social networks [16,17,18]. In addition, IA predisposes adolescents to higher levels of depression, suicidal ideation [19], and suicidal attempts due to the development of brain dysfunction: increased activity in the gyrus frontalis inferior of the right pars triangularis and the right pars opercularis [20]. These effects in young age can be quite alarming given the high prevalence of IA among adolescents (12–18 years old) in different parts of the world: up to 19.1% of Hong Kong Chinese adolescents, 18.8% of Taiwanese high school students, 11% of Greek adolescents, 11.6% of Turkish adolescents, 36.7% of Italian adolescents, 18.2% of Chinese junior high school students, and 38% of Korean adolescents (reviewed in [21]). IA prevalence among university students is also high, albeit a bit lower than that noticed in school children: 6.4% of first-year Chinese university students and 12.3% of Taiwanese university students (reviewed in [21]). Among adolescents, the literature confirms positive association of IA with attention deficit/hyperactivity disorder [22], mood disorders [23], social anxiety [21,24], hostility, and multiple addictions (e.g., smoking, binge drinking, and illicit drugs) [21,23,24]. Despite its widespread and serious adverse effects, IA may not be detected by health professionals due to lack of training [25].

The Internet Addiction Test (IAT), developed by Kimberly Young, is one of the most widely used measures to diagnose IA [7,26]. The IAT comprises numerous indicators of recurrent addiction behaviors including attributes of obsessive use of the internet (e.g., escapism, compulsivity, and dependency) and consequences of addictive use (e.g., personal and social conflicts, and personal and occupational performance deficiency) [27]. The IAT has been adapted to evaluate online sexual activities [28], and a modified version is utilized to assess smartphone addiction [29].

The IAT has been translated and validated into more than 20 languages other than English [25], including French [28,29,30], Spanish [31], Finnish [32], German [33,34], Italian [26,35], Polish [36], Turkish [37], Arabic [27,38], Greek [2], Romanian [6], Hebrew [39], Chinese [40], Indonesian [41], Malay [7], etc. However, there is less agreement on its construct structure. Several studies report a unidimensional factor structure of the IAT [26,29,30,32,37], including smartphone IAT [29]. Meanwhile, in some studies, the one-factor structure displays poor fit compared with a two-factor structure [6,28,31,33,34,35,36]. In few instances, the one-factor structure comparably fits data same as the alternative two-factor model [26,39]. A bifactor structure is reported to account for the high correlations between two extracted factors [5]. Some other studies reported a three-factor structure [40,41], four factor structure [18], or even five- or six-factor structures [7,42]. Interestingly, one study reported better fit for a one-factor solution based on all 20 items of the IAT than a well-fitting three-factor solution based on 18 items (after excluding item 5 and item 7) [41]. In some studies, the best fitting models were produced by removing up to three items [18,31,39,41] while some studies reported fitting models based on fit indices expressing values out of the acceptable range [6,35].

Variations in the structure of the IAT can be accounted, in part, by the method of extraction. Obviously, studies reporting more than two-factor structures used exploratory factor analysis (EFA) or principal component analysis [7,41,43], which employ methods that overestimate the number of extracted factors counting mainly on the criteria of eigenvalues > 1 [33,44,45]. However, studies using EFA in combination with more robust methods such as confirmatory factor analysis (CFA) report consistency of results of EFA with CFA indicators of the best fitting structures of the IAT [6,30,35,41]. CFA also uncovered failure of several items to contribute to any underlying latent factor structure [18,31,34,39,41], necessitating the need to revise the item structure of the IAT. In addition, the IAT is largely validated among adolescents, university students, and healthy young adults [25,41,43]. IA is high among individuals with comorbidities [23,24]. Therefore, the diagnostic potential of the IAT needs to be explored in diverse groups, including diseased conditions [25].

A few studies have evaluated invariance of the IAT across different groups such as gender [6] and Asian countries (Hong Kong, Japan, and Malaysia) [46]. Although in those studies the IAT was invariant, the latent factor structure of the IAT varied according to daily duration of internet use, online gaming, and young age in a French study [30]. Another study reported lower sensitivity of Chinese students to items 18, 19, and 20 than Malay students [18]. Gender, time spent online, internet use activities, perceived negative effect of internet use on academic performance and physical health, and years of internet use experience significantly affected the latent factor structure of the IAT [18].

Measurement invariance implies psychometric equivalence of a construct across groups, i.e., it has the same meaning to those groups [44,47]. Several forms of invariance are assessed in psychometric studies: (1) configural invariance—examines global model fit (the same number of factors is produced in all groups) without imposing constrains across groups, (2) metric invariance—examines the sensitivity of groups to each item on the measure by constraining factor loadings to equality across groups, (3) scalar invariance—examines variations in true mean differences by constraining intercepts of the regression equations of the observed variables on the latent factors to equality across groups, and (4) strict invariance—examines the uniqueness of each observed variable by constraining residuals to equality across groups [44,48]. Among all types, scalar variance/non-invariance is the most important because it may cause serious misinterpretation of true mean differences. One-third of the psychometric tests exhibits partial invariance [47]. On the other hand, strict invariance is rarely achieved; and it is not seriously considered in most instances [44]. Lack of evaluation of invariance of the IAT may cast doubt on the statistically significant differences in IA across different groups [47]. Therefore, establishing measurement invariance of the IAT in diverse groups is necessary for cross-group comparisons of mean differences and other structural parameters of IA [48].

Eating disorders (EDs) commonly develop in young groups, especially among adolescent girls and young women [49]. EDs are associated with high levels of emotional distress, which may trigger emotional eating and related obesity (e.g., in patients with bulimia nervosa, binge EDs, and food addiction) or excessive dieting and underweight in patients with anorexia nervosa (AN) in effort to control body weight and shape [50,51,52,53]. The COVID-19 outbreak has been associated with increased symptoms of dietary restriction, binge eating, purging, weight gain, and exercise behaviors in people with EDs and in the general population [54,55]. This is especially because individuals with EDs express difficulties with emotional regulation, which is associated with increased use of deficient ways of coping with the current lockdown. As a result, their psychopathology (depression and anxiety) heightens, which furthers emotional eating and eating pathology [54,56,57]. The dynamics of psychopathology in EDs varies by ED subtypes, entailing underweight, nutritional deficiency (e.g., tryptophan and micronutrients), and body dissatisfaction as causes of depression in the AN group [58,59]. On the other hand, depression is largely caused by neuroinflammation triggered by adipokines produced by body fatty tissues in ED subtypes characterized by overeating [60,61,62].

The state of being a student may be associated with distress due to academic demands; concerns about supporting tuition, housing, and other essentials for living; as well as concerns about attaining proper employment in the future [44]. Apart from other internet use activities, students also use the internet often to obtain resources necessary to support their education. However, excessive internet use causes distraction and evokes several problems—poor sleep quality, low self-esteem, social distress, low perceived social support, and poor communication skills [63]. Systematic meta-analyses show that IA is more common among students than adults working in the same field of study, e.g., medical and nursing students vs. health care professionals [63,64]. Metanalytic data also confirm that IA is more common among students with EDs than students with less ED symptomatology. In fact, IA among students is a significant predictor of different subtypes of EDs: AN, bulimia nervosa, binge-eating disorder, food preoccupation, loss of control eating, and dieting [65]. Nonetheless, IA among university students correlates more with symptoms of bulimia and binge eating than symptoms of AN [66,67]. EDs characterized by overeating display an addictive nature [49]. Hippocampal and insular levels of dopamine can be manipulated by certain foods (e.g., sugary and processed) leading to loss of control over the intake of these foods along with symptoms of craving and withdrawal [50]. Indeed, people with EDs have a high genetic tendency toward addictive and impulse control disorders, which justifies the high comorbidity of these diseases [52,68,69].

Prolonged and repeated internet use is associated with reduced physical activity and increased consumption of fast food because of shrinkage of time allowed for cooking healthy food (containing fibers), which favor frequent snacking, food preoccupation, and loss of control eating [65,70,71]. IA, including smartphone addiction, as well as the duration of internet use, is positively associated with body mass index [49,66]. Body mass index interacts with the sociodemographic and clinical variables to affect addictive behaviors (e.g., smoking and IA) differently in ED subtypes characterized by excessive dieting or overeating [49,72]. Although less common, dieting behaviors among students with IA is probably triggered by following famous people on social networking sites, which is commonly associated with comparing one’s shape with that of the influential model. This can largely affect self-perception and mood, resulting in excessive dieting in response to individuals’ desire to resemble their influential models [65].

Women with EDs frequently join online self-help groups and digital ED interventions, and those with higher addictive tendencies may develop IA [49,72]. Given the innate nature of distress in EDs, IA may be an additional source of distress that may promote a prolonged course of the disease and threaten patients’ wellbeing and quality of life [67,72]. Therefore, careful identification and proper management of IA in patients with EDs may have implications for improving their recovery. This may not be currently applicable because of the lack of calibrated measures of IA among pathological conditions such as EDs. Although IA is widespread among adolescents and university students, and it is associated with psychopathology, including EDs [21,65]; it is not clear whether IA conceptualization varies among clinically diagnosed patients with EDs and those without a formal diagnosis.

To fill the gap, the current study aimed to examine the construct structure and invariance of the IAT among healthy school children, women with EDs, and university students. The structure of the IAT has been extensively revised among women with EDs based on item loadings, item-total correlations, and Griffith’s addiction component model. The resulting short version was examined for invariance across ED subtypes in the same sample. Because IA is reported to correlate with all ED subtypes [65], we hypothesized that the structure of all the IAT versions would be invariant across ED subtypes. The six-item IAT was also tested among healthy university students for its construct validity and invariance across groups of gender, nationality, academic grade, major, and internet use activities. Because the six-item IAT was supposed to eliminate most irrelevant items of the parent scale, we hypothesized that it would be invariant across different student characteristics. Additionally, it was examined for invariance across groups of females (women with EDs vs. healthy students). Again, we hypothesized that the short version of the IAT may express invariance across healthy and ED females. To evaluate the criterion validity of the IAT and its short version, we hypothesized that ED patients with higher IAT scores would express higher Facebook dependence. Likewise, we hypothesized that higher IAT scores would strongly correlate with perceived negative effects of internet use on academic performance and other IA outcomes among university students.

## 2. Methods

This study evaluated the structure and invariance of the IAT across healthy and diseased groups from different cultural backgrounds. The current analyses integrated publicly accessible datasets involving a sample of school children from Hong Kong [73], a sample of women with EDs from Spain [74], and a sample of Malay/Chinese university students from Malaysia [75]. These datasets are affiliated with three previously published cross-sectional studies, which have attained ethical approval for their data collection protocols [18,72,76]. Therefore, we have not obtained ethical approval for the present study. Three studies based on these discrete samples were conducted to (1) examine the factor structure of the IAT using EFA (study 1) and CFA (study 2); (2) develop a briefer version of the IAT (six-item IAT) and examine its invariance across ED groups (study 2); (3) examine invariance of the six-item IAT across different groups of students as well as across female students and women with EDs (study 3); and (4) examine the criterion validity of all versions of the IAT (study 2 and study 3)

### 2.1. Study 1

#### 2.1.1. Study Design, Participants, and Procedure

This study comprised a clustered random sample of school children, herein referred to as the “school children sample”, which was obtained from nine schools in Hong Kong during the period between May and July 2017. Out of 1121 pupils attending on the survey date, whose parents had previously signed an informed consent, 1097 pupils (97.9%) agreed to participate—they completed an assent form. The authors of the original dataset had removed twenty-five responses because they were outliers [76]. The final sample (*N* = 1072, mean age = 12.4 ± 2.0 years, males = 62.9%) comprised children from elementary (*N* = 462), junior (*N* = 357) and high schools (*N* = 235). The Human Research Ethics Committee of the Education University of Hong Kong approved the data collection protocol [76].

#### 2.1.2. Data Collection Measures

Problematic use of the internet was assessed by the IAT, a 20-item measure with responses rated on a five-point Likert scale (0 = not applicable, 1 = never, 2 = rarely, 3 = sometimes, 4 = often, and 5 = very often) [33]. In general, items of the IAT reflect internet use habits (e.g., form new relationships with fellow online users (item 4) and check your email (item 7)); aspects of internet dependence (e.g., become defensive or secretive (item 9), fear that life without the internet would be boring (item 12), and hide how long you’ve been online (item 18)); preoccupation with Internet use (e.g., feel preoccupied with the Internet when off-line (item 15)); compulsivity (e.g., stay online longer than you intended (item 1) and choose to spend more time online (item 19)); escapism (block out disturbing thoughts about your life with soothing thoughts of the Internet (item 10)); as well as problems in everyday functioning and other drawbacks of internet addiction (neglect household chores (item 2), lose sleep (item 14), and your job performance or productivity suffers (item 8)) [27,35]. The IAT scores range between 0 and 100 [25,33]. Scores below 30 indicate normal use, and higher scores indicate IA, which can be mild (31–49), moderate (50–79), and severe (80–100) [25]. Its reliability in this sample is excellent (Cronbach’s alpha = 0.99).

#### 2.1.3. Statistical Analysis 

In SPSS, age was described by mean ± SD while numbers and percentages were used to describe other characteristics of the sample. EFA with maximum-likelihood extraction and varimax rotation was used as an initial step of evaluating the structure of the IAT as noted by the number of factors with eigen values > 1. This is because it allows items to load freely on the corresponding factors without implying any constraints. To ensure that the sample size is adequate for EFA, the analysis included Kaiser-Meyer-Olkin (KMO) measure of sampling adequacy and Bartlett’s test of sphericity. Significance was considered at a probability of 0.05, two-tailed.

### 2.2. Study 2

#### 2.2.1. Study Design, Participants, and Procedure

This study is based on a convenient sample (*N* = 123) of Spanish females (mean age = 27.3 ± 10.6 years) consecutively treated for EDs at the outpatient and inpatient departments of the General University Hospital of Ciudad Real between February and November 2018. Participants were included if they were >12 years of age and agreed to sign an informed consent. For patients below 18 years, their guardians signed informed consent. Patients with physical or mental disabilities were excluded. Reported diagnoses of EDs were AN (*N* = 59, 48.0%), bulimia nervosa (*N* = 35, 28.5%), binge EDs (*N* = 11, 8.9%), and EDs not otherwise specified (*N* = 18, 14.6%). On average, participants had a diagnosis of EDs for 10.2 ± 8.0 years, and they have been in treatment for an average of 8.1 ± 6.5 years. The average body mass index was 22.2 ± 8.4, with a significant difference between patients with AN (group1) and patients with other EDs (group2) t(73.0) = −6.77, *p* = 0.001. Around half the participants (*N* = 63, 51.2%) had a history of psychiatric hospitalization while 48.8% had never been hospitalized. Less than half the participants (*N* = 56, 45.5%) reported tobacco smoking. Depression was the only reported psychiatric comorbidity; it co-occurred in 29.3% of the participants. Most participants were single (*N* =100, 81.3%). Few participants had only elementary school education (*N*= 5, 4.1%), more than half the sample (*N* = 78, 63.4%) had high school education, and the rest of the participants had a university degree. The Ethics and Clinical Research Committee of Ciudad Real (Spain) approved data collection (ref. 2017C/123) [72].

#### 2.2.2. Data Collection Measures

Data were collected from eligible participants through a self-administered questionnaire that comprised a section inquiring about participants’ sociodemographic and clinical characteristics such as age, weight, height, ED subtypes, treatment history, comorbidities, and smoking status [72].

The Spanish version of the IAT (Appendix A), which has been validated among university students [31], was used to assess problematic internet use. Its reliability in the current study is excellent (Cronbach’s alpha = 0.99).

The validated Spanish version of the Bergen Facebook Addiction Scale (BFAS) [77] was used to evaluate Facebook dependence. The BFAS comprises six items, which measure different aspects of Facebook addiction. Items are rated on a five-point Likert scale (1 = very rarely to 5 = very often). Scores higher than 18 indicate higher levels of addiction [3]. The BFAS is reported to exhibit sound psychometric properties in this ED sample, including excellent internal consistency (Cronbach’s alpha = 0.99) [78]. Scores of the BFAS were used to test for criterion validity.

#### 2.2.3. Statistical Analysis

The dataset was checked for missing data, and one incomplete response (data on psychiatric comorbidity) was deleted, resulting in a final sample size of 123. The normality of the IAT was examined by Shapiro-Wilk and Kolmogorov-Smirnov tests of skewness and kurtosis. Because AN was the most dominant diagnosis and other EDs were less presented, we created two major categories of EDs for subgroup analysis: patients with AN (group 1) and patients with other EDs (group 2). These two groups were established based on the fact that body mass index was significantly lower in group 1 than in group 2 [49].

In a second stage, we used CFA with maximum likelihood method and bootstrap based on 2000 random samples and a 95% confidence interval (95% CI) to examine different structures of the IAT commonly reported in previous studies [31,33,34,43], including the 12-item IAT proposed by Pawlikowski et al. (2013) [33] (Table 1). Because item 7 was reported to be problematic in several studies, including those evaluating the IAT among Spanish students [31], and it had very low loading in our school children sample, models were tested both with and without item 7. We extensively revised the IAT based on Griffiths’ addiction components model [13,14]. From the 20 items of the IAT, we retained only six items (Appendix A)—those with the highest loadings that seemed to be most relevant to the six components covered by Griffiths’ addiction components model.

A non-significant chi square index (χ^2^) was used to reflect global data-fit to the models [45]. Model-data fit was considered good and acceptable, in order, if values of the Comparative Fit Index (CFI) and the Tucker–Lewis Index (TLI) were equal to or above 0.95 and 0.90, respectively, along with values of the root mean square error of approximation (RMSEA) and standardized root-mean-square residual (SRMR) less than 0.06 and 0.08 [31,44]. This combination facilitates a sound judgement because some indices such as RMSEA and SRMR may be sensitive to misspecification of the factor loadings or misspecification of the co-variances, respectively [31,79]. Based on modification indices, all crude models of the IAT were modified by allowing some residuals to correlate.

Multigroup analysis was conducted to examine if models expressing satisfactory fit (Model 8, Model 11, Model 17, and Model 19) are invariant across ED groups. Invariance was evaluated at the configural, metric, scalar, and strict levels as described above [44,48]. To identify items that represent the source of invariance, we created several models in which the parameter estimates of each item were constrained to equality between groups while the rest of the items were left to load freely between the two groups. In nested model comparisons, significant χ^2^ highlighted a significant decrease in model fit when path coefficients were constrained to equality between groups [47]. Critical ratios for differences between parameters with values out of the range of −1.96 and 1.96 pinpointed variant items.

Reliability of the IAT and the six-item IAT was evaluated by Cronbach’s alpha [80], corrected item-total correlations, and alpha-if-item-deleted. Item coverage and predictive validity of the six-item IAT were evaluated by correlating its score with the original 20-item IAT in all ED samples. Criterion validity was tested by Spearman’s rho correlation between scores of the IAT and the BFAS. The analyses were conducted in SPSS and Amos version 24, and significance was considered at a probability of 0.05, two-tailed.

### 2.3. Study 3

#### 2.3.1. Study Design, Participants, and Procedure

This study included a purposive sample of undergraduate students (*N* = 1120, mean age = 21.1 ± 1.6) from both sexes and all grades obtained from Univerisiti Teknologi Malaysia (UTM). A detailed description of the sociodemographic and internet use characteristics of the participants is presented in Table 2. All procedures performed in this study involving human participants were in accordance with ethical standards of institutional and national research committee and with the 1964 Helsinki declaration and its later amendments or comparable ethical standards. All participants signed a written consent form [18].

#### 2.3.2. Data Collection Measures

Data were collected through a pencil-paper questionnaire involving the IAT, along with questions about the sociodemographic (e.g., age, gender, nationality), academic (e.g., major, academic grade), and internet use data (e.g., duration of time spent online, activities of internet use such as gaming, social networking, general use such as shopping, etc.), and the effect of internet use on academic performance, which was assessed by a single question: rate the extent to which internet use affects your academic performance on a scale from 1 (no effect at all) to 5 (extremely negative effect).

#### 2.3.3. Statistical Analysis

One response was removed from the dataset because of missing data (the IAT). Based on study 2, we used CFA to examine the structure Pawlikowski et al. (2013)’s 12-item IAT [33] (Model 17) and our six-item (Model 19) in UTM students. We examined the invariance of both models across groups of gender, academic grade, major, nationality, and internet use activity (group categories are shown in Table 2). In addition, we examined invariance of these models across groups of Spanish women with EDs and healthy UTM female students. The Model fit in CFA and multigroup CFA analysis is based on the criteria described above in study 2.

To examine criterion validity in this sample, we used Spearman’s rho test to correlate the scores of the 12-item IAT and the six-item IAT with time spent online, years of internet use experience, and perceived effect of internet use on academic performance.

## 3. Results

### 3.1. Study 1

Exploratory Factor Analysis of the Internet Addiction Test

Four factors with eigen values >1 were extracted by EFA. They explained 32.6%, 6.8%, 5.7%, and 5.1% of the variance in the IAT. The values of KMO test (0.935) and Bartlett’s test (χ^2^(190) = 6175.11, *p* < 0.001) indicated adequacy of the sample size and appropriateness of participant-to-item ratio for EFA. While items 5, 7, and 9 had loadings below 0.3, several items significantly loaded on two or three factors (Table 3). Item communalities, scree plots, and reproduced correlations are presented in Appendix A.

### 3.2. Study 2

#### 3.2.1. Confirmatory Factor Analysis of the Internet Addiction Test

As shown in Appendix A, all the crude models (Model 1, Model 3, Model 5, Model 7, Model 9, and Model 12) had poor fit in all the samples except for Model 3 and Model 7, which had CFI within the acceptable range mostly in group 2. Allowing the residuals of some items to correlate improved the data-model fit in the whole sample and in group 2, albeit RMSEA was exceptionally high. Noticeably, items with correlating residuals in most models were (items 8 and 9), (items 14 and 15), (items 15 and 16), and (items 17 and 18). Meanwhile, the data fit for all models tested in group 1 was unsatisfactory. However, CFI values in Model 8 and Model 11—which tested the two-factor structures proposed by Fernández-Villa and Barke (for 19 items after removing item 7)—were within the acceptable range (0.903 and 0.901, respectively), indicating some acceptable fit of these two models. The fit of a second order factor was the same as the two-factor structures. A bifactor structure based on Model 8 had poor fit (Appendix A), but a bifactor structure based on Model 11 had the best fit in all ED sample, Model 15.

As shown in Appendix A, the unidimensional structure of the six-item IAT expressed the best fit in all ED samples. For the first time, RMSEA considerably decreased, and χ^2^ was not significant only in Model 19 when error residuals were correlated (Figure 1), which denotes that this model describes the best structure of the IAT.

#### 3.2.2. Invariance of the Internet Addiction Test across Eating Disorders

To determine whether different IAT structures vary across ED groups, we conducted multigroup analysis to compare data fit to the one-factor and two-factor structures of the IAT both with and without item 7 (Model 2, Model 4, Model 6, Model 8, and Model 11). All these models held configural invariance among ED groups, but did not hold scalar and metric invariance, except for Model 11 (Table 4). Examinations of each individual path revealed that items 14, 15, and 16 were the source of invariance in the tested models (more details in Appendix A). In particular, women with AN seemed to be less sensitive to these items than women with other EDs. For the 20-item IAT, Model 11 was the only model holding configural and metric invariance across patient groups. However, it expressed scalar variance due to variations in the mean differences in the shared variances of both factors and inter-factor correlations. Sources of invariance in Model 17 (the 12-item IAT) involved variations in the loading of item 15 and the shared variance in item 14 across groups, in addition to variations in the inter-factor correlation and the correlations between the error terms of item 14 with item15 and item 17 (Appendix A). Model 19 (the six-item IAT) held configural and metric invariance but did not hold scalar and strict invariance due to variations in the mean differences in the shared variance of item 17 and the extracted overall six-item factor (Appendix A).

#### 3.2.3. Reliability and Criterion Validity of the Internet Addiction Test

The IAT, the 12-item IAT, and the six-item IAT demonstrated excellent reliability in all the samples as noted by high values of coefficient alpha (Table 5). All corrected item-total correlations and values of alpha if-item-deleted were remarkably high, indicating that all items considerably contribute to the overall latent construct under measurement, which reflects good convergent validity of these measures. The total scores of the three versions of the IAT strongly correlated with problematic use of Facebook in all ED samples (Table 5) denoting adequate criterion validity. The shortened versions of the IAT strongly correlated with the original IAT pinpointing their adequate predictive validity. The normality of the 12-item IAT and the six-item IAT is comparable with that of the original IAT, as noted by values of the Shapiro Wilk test (Table 5) and Kolmogorov-Smirnov test (Appendix A).

### 3.3. Study 3

#### 3.3.1. Factor Structure of the Internet Addiction Test

CFA involving correlating few error terms (Figure 2) revealed acceptable fit of the factor structure of the 12-item IAT and good fit of the factor structure of the six-item IAT (Table 6).

#### 3.3.2. Invariance of the Internet Addiction Test

The results of multigroup CFA in the student sample uncovered invariance of the 12-item IAT and the six-item IAT at the configural, metric, scalar, and strict levels across groups of gender, nationality, major, academic grade, and types of internet use activities (Appendix A).

Invariance analysis across groups of female students and women with different EDs shows configural invariance in all models (Table 7 and Table 8). However, both models exhibited non-invariance at the metric and scalar levels when the model fit was compared across female students and all women with EDs or women with other EDs. Nonetheless, models compared across female students and women with AN held invariance at the scalar level but not at the strict level.

#### 3.3.3. Reliability and Criterion Validity of the Internet Addiction Test

The reliability of the IAT and the 12-item IAT was very good while that of the six-item IAT was good (Table 9). The total scores of the three versions of the IAT strongly correlated with time spent online, years of internet use experience, and perceived effects of internet use on academic performance (Table 9). The shortened versions of the IAT strongly correlated with the original scale and their normality was comparable with that of the original IAT (Table 9, Appendix A).

## 4. Discussion

Given that the psychometric properties of the IAT have always been tested among students (in 20 out of 25 studies) and healthy young adults [25], the current study enriches the existing knowledge by reporting on its structure and invariance among both healthy youth and patients with EDs. In addition to examining data fit to several structures of the IAT via numerous robust validation techniques, the study represents the first attempt to extensively revise this measure based on Griffiths’ addiction components model denoting usability and comparable or even superior psychometric properties of the six-item IAT to those of the original scale.

Our results suggest impure factorial structure of the IAT. EFA in the school children sample uncovered the presence of four factors, with three items failing to load on any factor and numerous items with high cross-loadings (Table 3). Likewise, in our university student sample, the IAT is reported to express three factors with items 3, 7, and 9 failing to load on any factor [18]. In a previous Spanish study, EFA indicated that the IAT had three factors among university students, with a single item loading on the third factor and item 7 not loading on any factor. Therefore, Fernández-Villa et al. deleted item 7 and reported the findings of EFA and CFA based on a two-factor solution. Nonetheless, both factors together explained 55% of the variance in total, and the reliability of the second factor was relatively poor (Kappa = 0.65) [31]. Unlike student samples, CFA in the ED samples revealed that all items of the IAT had high loadings on a single factor (0.86–0.93) or two factors (0.84–0.94; Appendix A), albeit suboptimal fit was flagged by several fit indices. Variations in the IAT structure in school/university students and ED patients denote that the number of variables, which should covey the construct of IA as covered by the IAT is not consistent in healthy and diseased individuals as we hypothesized.

Although items 5, 7, and 9 did not load on any factor in the school children sample, they were not a source of noise in the structure of the IAT in the ED sample (see CFA models in Appendix A). Nonetheless, item 7 caused misfit in a sample of Spanish university students [31] as well as in German [33], Romanian [6], and Indonesian [41] students, as well as German young adults [33]. Therefore, we ran all CFA models in study 2 with all the 20 items and without item 7. Remarkably, the 20-item one-factor structure with correlating errors (Model 2) exhibited a slightly better fit than the 19-items one-factor structure with correlating errors (Model 4) in the ED samples, but its fit in group 1 was poor (Appendix A). Interestingly, the best fit in all ED samples was exhibited by two-factor structures of a 19-item IAT, Model 8 (Emotional Investment; Performance and Time Management) [31] and Model 11 (Emotional and cognitive preoccupation; Loss of control and interference with daily life) [34], along with Model 15—a bifactor structure involving a general IA factor with two specific factors (Emotional and cognitive preoccupation; Loss of control and interference with daily life). In fact, they were the only models of the full IAT with acceptable fit in all ED samples, particularly women with AN.

It is not clear why removing item 7 improved the fit in these structures and in women with AN, in particular. However, other studies reporting issues associated with item 7 denote that it may be an outdated item [6,31,33] because the IAT was developed more than two decades before, and e-mail represents a fast and useful tool for communication nowadays. Thus, checking the e-mail box frequently may be a normal behavior or a routine of everyday life (e.g., for professional reasons). Within this context, addiction may be better reflected by excessive time used for checking e-mail rather than the frequency of e-mail checking [6].

Evidence on invariance of the IAT is limited. The Romanian two-factor structure of the IAT expressed configural, metric, and scalar invariance across gender groups, albeit overall fit of the model was fairly low [6]. Variance of the IAT across gender and nationalities has been previously reported among university students in Malaysia [18]. Across ED groups, all IAT structures demonstrated configural invariance, denoting that women in different ED groups can conceptualize problematic use of the internet similarly. Nonetheless, women’s response to items 14 “lose sleep due to late night log-ins”, 15 “feel preoccupied with the internet when off-line or fantasize about being online”, and 16 “find yourself saying just a few more minutes when online” in group 1 was less than in group 2 for all models, except for Model 11, which demonstrated scalar variance—expressed by differences in the shared variance of both factors and inter-factor correlation between groups (Appendix A). Of interest, correlating the residuals of item 15 with the two other items improved fit indices in most models of the 20-item IAT. Meanwhile, when Model 15 was compared across ED groups, it failed to converge; most items with correlating residuals expressed standard errors > 1. Increasing the iteration limit and removing these items failed to support model convergence and resulted in more dysfunctional items. In line with our results, item 15 caused misfit in the Hebrew IAT, and its removal along with item 12 achieved an acceptable fit of the two-factor structure [39]. Altogether, variance of the IAT across ED groups is inconsistent with our hypothesis. Meanwhile, correlating and variant items on the IAT may threaten the validity of its measurement.

Pawlikowski and colleagues developed the 12-item IAT by removing items with low loadings [33]. Although the 12-item IAT expressed a better fit than the 20-item IAT in our ED samples, χ^2^ measure of global model fit and RMSEA measure of absolute model fit [47] denoted unsatisfactory fit. In addition, the model exhibited between group variance involving item loadings, item residuals, and the variance explained by factors underlying the scale (Table 4, Appendix A). Cumulative knowledge highlights the importance of combining statistical and content approaches in composite scale reduction to ensure that the shortened version retains the validity and other psychometric properties of the original scale [79,81]. Accordingly, we have extensively revised the IAT by stepwise including a single item (the most relevant with the highest loadings) to represent each of the six components on Griffiths’ addiction components model, ending with a six-item version that evaluates conflict, mood modification, salience, tolerance, withdrawal, and relapse components of IA (Appendix A). 

This six-item IAT expressed the best fit in all ED samples, with less variance across ED groups compared with the 20-item and 12-item versions. Among university students, the six-item IAT expressed better fit than the 12-item IAT (Table 6). As hypothesized, the six-item IAT was invariant at all levels across groups of gender, nationality, major, academic grade, and internet use activity (Appendix A). On the contrary, a former investigation involving the current university student sample reports best fit for a four-factor model of the IAT, which comprised only 17 items because items 3, 7, and 9 were removed due to suboptimal loadings [18]. Nonetheless, that 17-item model expressed variance across nationalities, with a lower tendency of Chinese students to endorse items 18, 19, and 20 than Malay students; females had a lower tendency to endorse item 14 than males; and students spending more time online were more likely to report higher scores on items 1 and 12 [18]. Thus, our findings support complete invariance of the six-item IAT across different groups among healthy respondents. Noticeably, invariance of the 12-item IAT and the six-item IAT at the scalar level across groups of Asian healthy female students and European female patients with AN (Table 7 and Table 8) emphasizes the ability of these measures to objectively depict IA in different cultures as well as in healthy and diseased individuals. Nonetheless, variance of all the IAT versions at the scalar level across AN and other EDs (Table 4) as well as across healthy university students and women with other EDs, primarily bulimia nervosa and binge EDs (Table 8), denote that excessive eating, rather than dieting, is likely to be associated with excessive internet use. This finding is inconsistent with our hypothesis, but it is consistent with a former study reporting higher levels of IA among students exhibiting symptoms of bulimia and binge eating [67]. Body mass index, which tends to increase in EDs characterized by excessive eating, is associated with IA among university students [66], and it predicted IA among ED women in group 2 in another investigation [49].

Invariance of the IAT among patients with different EDs may be interpreted within the context of the discrete physiological and symptomatic features relevant to increased addictive tendencies in specific ED subtypes. Food intake in EDs is associated with dysregulations in neurotransmitters, neuropeptides, and peripheral peptides involved in emotional and reward pathways that regulate eating behaviors [82]. Cumulative knowledge shows that serotonin, dopamine, and prostaglandin promote feeding behaviors while neuropeptide Y, norepinephrine, GABA, and opioid peptides inhibit feeding [83]. In experimental models of binge eating, dysregulation of the endocannabinoid system (which regulates cognition, emotions, and reward response in additive disorders) was depicted by selective down-regulation of fatty acid amide hydrolase gene with a consistent reduction in histone 3 acetylation at lysine 4 of the gene promoter only in the hypothalamus region of the brain [84]. In another experiment, rats with hyperphagia and obesity induced by high-fat and high-sugar diets expressed dysregulation of dopamine and endocannabinoid system gene expression in reward and homeostatic brain regions. CB2 receptor mRNA expression increased in the nucleus accumbens of rats on high-sugar diet while CB1 receptor mRNA expression decreased in obesity-prone rats [85]. Indeed, EDs characterized by compulsive overeating (e.g., binge EDs and bulimia nervosa) are considered a phenotype of addictive disorders [86]. Neurotransmitters known to inhibit eating behaviors such as norepinephrine are deficient in adolescents with IA, and they correlate with symptoms of depression and anxiety in those adolescents compared with healthy controls [87]. Therefore, signal transduction alterations conducive to addictive tendencies in EDs characterized by excessive overeating as well as metabolic and inflammatory alterations resulting from their eating behavior (e.g., obesity) [88] may increase the risk for IA in those patients. This notion is suggested by higher endorsement of item 15 and item 16, which reflect the salience and tolerance components of IA.

A systematic review comprising 23 studies associates IA with decreased sleep duration (odds ratio = −0.24, 95% CI: −0.38 to −0.10) and increased incidence of sleep problems (odds ratio = 2.20, 95% CI: 1.77 to 2.74) [89]. Therefore, it may be intuitive that patients with ED subtypes that are prone to IA express sleep loss due to late night log in, which is noted by item 14. In fact, nocturnal eating behavior and sleep-related eating disorder-like behavior, which are associated with overeating at night have features similar to bulimic and binge subtypes of EDs [90]. Nocturnal eating may coincide with late night log in and subsequent sleep loss. Our results, along with reports from the available literature, denote a higher tendency for IA among women with ED subtypes characterized by compulsive eating and high body mass index [49,66]. Accordingly, the management of IA in this patient group may facilitate recovery by alleviating dual comorbidities (IA, EDs, obesity, and psychopathologies, e.g., sleep and mood dysfunctions). Further investigations are needed to explore the dynamics underlying IA in different subtypes of EDs.

Apart from its superior fit and invariance, the six-item IAT enjoys additional excellent psychometric properties. Because multiple replicate items contribute to scale reliability, eliminating items is usually associated with a decrease in scale reliability unless it involves heterogeneous items [79,91]. Despite the extensive reduction in its number of items, the internal consistency of the six-item IAT was excellent and good in the ED and university student samples (Table 5 and Table 9). The normality of the six-item IAT is also congruent with that of the IAT and the 12-item IAT in both samples. Unlike the IAT and the 12-item IAT, all item-total-correlations on the six-item IAT were high (Table 5 and Table 9), noting its high convergent validity. Thus, this brief six-item version retains items most relevant to the construct of IA, i.e., its items possess the highest sensitivity and specificity [91]. This gets further support from criterion validity tests, which show that the six-item IAT correlated with perceived effect of the internet on academic performance at a higher level of significance than that expressed by the parent scale (Table 9). It also correlated with Facebook addiction in ED patients at the same level as the parent scale. These results support our hypothesis concerning the use of the six-item IAT as a valid criterion that can strongly correlate with other relevant constructs. Its high correlation with the original IAT pinpoints its adequate item coverage and strong predictive validity. Taken together, the six-item IAT as a measure of IA may be superior in its psychometrics to all versions of the IAT. An exceptional merit of the six-item IAT is its brevity, which permits health professionals to rapidly screen for IA and to include additional measures on test batteries for comprehensive assessments. More investigations are necessary to examine invariance of these measures across other groups.

This study has the merit of being the first to examine the IAT in a clinical sample and to develop and test various psychometric properties of the six-item IAT in culturally-diverse clinical and healthy samples. Shorter versions of the IAT were invariant across Malay and Chinese students, who both live in the same part of the world—south-east Asia. However, examining invariance of these short versions of the IAT across Asian healthy female students and European women with EDs may not be standardized enough to judge invariance of the 12-item/six-item IAT in these populations. Therefore, future studies are encouraged to evaluate the effect of cultural variations on the properties of the six-item IAT both among healthy and diseased groups. This study has also other limitations. Test-retest reliability is important to present the psychometric soundness of a new scale; however, it was not possible to conduct that test, given the nature of the used data. In all the samples, IA was not diagnosed by a clinician based on the diagnostic criteria found in the DSM-5. Thus, we were not able to examine whether the six-item IAT could differentiate between healthy individuals and those with an internet use disorder. 

Because this research is based on public data, some important details on how the research was conducted in the original study are lacking. For example, the recruitment method and specific sampling strategy taken to obtain the university student sample are not described [18], which may imply risk for selection bias. Response rate for surveys, priori power analysis, as well as the number of individuals excluded for not meeting the inclusion criteria were lacking in the three studies. We have not explored cut-off points for categorizing IA based on the six-item IAT. Nonetheless, future investigations may explore the usefulness of cut-off points based on a polythetic scoring scheme (scoring 3 or above on at least four of the six items) or a monothetic scoring scheme (scoring 3 or above on all six items), i.e., total scores of 12 or 18 may reflect IA based on liberal and conservative approaches of categorization [3,78]. 

Although the six-item IAT was invariant across gender among university students, we could not examine invariance across gender in the ED sample because it comprised females only. Invariance of the IAT across gender groups may not hold among ED patient population. This is because EDs are more frequent and more comorbid among women [92] who also express higher psychological distress and IA than men [23,93] while research confirms differences in brain activity between men and women with internet gaming disorder [94]. Additionally, subtypes of EDs (e.g., bulimia nervosa, binge eating, etc.) were not detailed in invariance analysis because they were less represented in the ED sample. Additionally, the ED sample size was not determined based on power analysis, entailing a need for further testing in larger samples. In addition, our clinical sample was limited to Spanish patients with EDs in a single hospital, which limits generalizability to other patient groups, facilities, and countries.

## 5. Conclusions

Using already existing data, this study is the first to examine the psychometric properties of the IAT in a clinical sample (women with EDs) and compare them across healthy students from different cultures. The new scientific knowledge attained in this study showing that the IAT expressed four-factor structures in school and university students, but it is best described by a bifactor structure or a two-factor structure among women with EDs. Both genders as well as healthy and ED women had the same global conceptualization of IA. However, excessive eating was associated with increased intensity of IA “tolerance, salience, and sleep loss”. Therefore, researchers developing interventions to address IA among patients with EDs should pay more attention to ED subtypes characterized by overeating, eating at night, and poor sleep. 

Extensive revision of the scale resulted in a unidimensional six-item IAT, which is capable of detecting the domains of IA according to Griffiths’ addiction component model. The six-item IAT expressed better psychometrics compared with the parent scale and the 12-item IAT. Thus, it may be reliably used for prompt capturing of IA in research and clinical practice, particularly during the current COVID-19 outbreak, which has witnessed an expansion in internet use and IA. Its brevity may support comprehensive assessments by allowing test batteries to include more measures, which may promote efforts for reducing comorbidities and enhancing recovery. To maximize the utility of this brief scale, more investigations of its psychometric properties in larger samples with more representation of ED subtypes as well as in more diverse clinical conditions and healthy groups from different cultures are needed.

## Figures and Tables

**Figure 1 ijerph-18-12341-f001:**
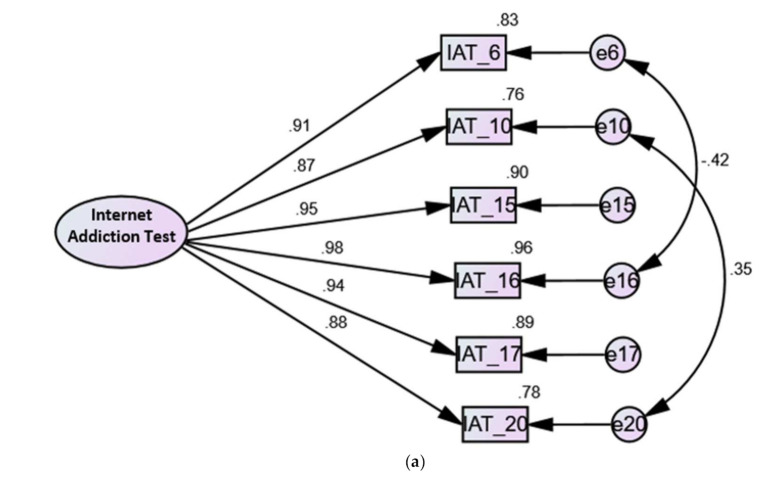
Factor structure of the six-item version of the Internet Addiction Test, Model 19, among all women with eating disorders (**a**), women with anorexia nervosa (**b**), and women with other eating disorders (**c**). IAT: Internet Addiction Test.

**Figure 2 ijerph-18-12341-f002:**
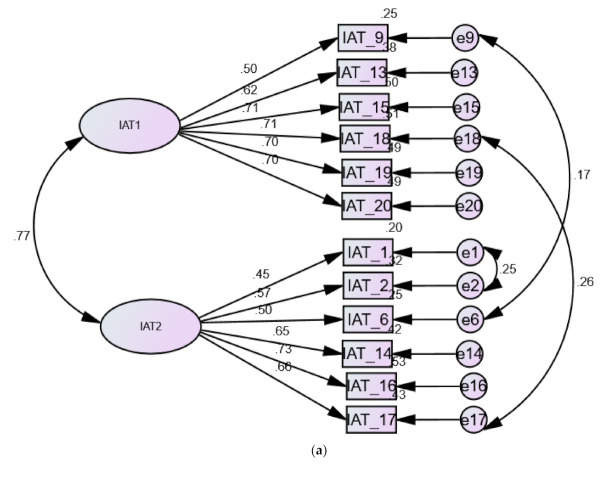
Models representing the 12-item IAT (**a**) and the six-item IAT (**b**) among university students.

**Table 1 ijerph-18-12341-t001:** Models proposed for the factor structure the Internet Addiction Test and items on each corresponding factor.

Factors	Items Comprising Factors in Each Model					
	Model 1 Model 2, Model 3 ○, Model 4 ○	Model 5, Model 6, Model 7 ○, Model 8 ○ (Fernández-Villa et al., 2015)	Model 9, Model 10, Model 11 ○, Model 15 ○◭ (Barke, et al., 2012)	Model 12, Model 13, Model 14 ○ (Widyanto, et al., 2011)	Model 16, Model 17 (Pawlikowski, et al, 2013)	Model 18, Model 19
Internet addiction	1–20					6, 10, 15, 16, 17, 20
Emotional investment		3, 4, 9, 10,11,12, 13, 14, 15, 19, 20				
Performance and time management		1, 2, 5, 6, 7, 8, 16, 17, 18				
Emotional and cognitive preoccupation			3, 4, 5, 9, 11, 12, 13, 15, 18, 19, 20			
Loss of control and interference with daily life			1, 2, 6, 7, 8, 10, 14, 16, 17			
Emotional/psychological conflict				3, 5, 8, 9, 10, 11, 17, 18, 19		
Time management				1, 2, 6, 7, 16		
Mood modification				4, 12, 13, 14, 15, 20		
Craving/social problems					9, 13, 15, 18, 19, 20	
Loss of control/Time management					1, 2, 6, 14, 16, 17	

○: Item 7 is removed from the model, ◭: a bifactor structure.

**Table 2 ijerph-18-12341-t002:** Sociodemographic, academic, and internet use characteristics of participants in the university student sample.

Participants’ Characteristics	All Students (*N* = 1119)	Males (*N* = 629)	Females (*N* = 490)
	No (%)	No (%)	No (%)
Age mean (SD) in years	21.1 (1.6)	21.0 (1.7)	21.2 (1.6)
**Nationality**			
Malay Chinese Others	723 (64.6) 321 (28.7) 75 (6.7)	402 (63.9) 180 (28.6) 47 (7.5)	321 (65.6) 141 (28.8) 28 (5.7)
**Academic major**			
Art, humanities and social science Science Engineering Others	136 (12.2) 377 (33.7) 523 (46.7) 83 (7.4)	53 (8.4) 150 (23.8) 393 (62.5) 33 (5.5)	83 (16.9) 227 (46.3) 130 (26.5) 50 (10.2)
**Academic grade**			
Freshman Sophomore Junior Senior	278 (24.8) 266 (23.8) 309 (27.6) 266 (23.8)	168 (26.7) 167 (26.6) 161 (25.6) 133 (21.1)	110 (22.4) 99 (20.2) 148 (30.2) 133 (27.1)
**Common internet use activities**			
Gaming Social networking General use Others	132 (11.8) 812 (72.6) 133 (11.9) 42 (3.8)	121 (19.2) 412 (65.5) 69 (11.0) 27 (4.3)	11 (2.2) 400 (81.6) 64 (13.1) 15 (3.1)
Time spent online	6.5 (4.9)	6.7 (5.1)	6.3 (4.6)
Perceived effect on study	3.4 (1.0)	3.3 (1.0)	3.5 (1.0)
Years of internet use experience	7.6 (3.0)	7.6 (2.9)	7.6 (3.0)
20-item IAT mean (SD)	48.1 (15.1)	49.4 (15.4)	46.3 (14.5)
12-item IAT mean (SD)	27.7 (10.0)	29.9 (10.2)	27.5 (9.7)
Six-item IAT mean (SD)	14.0 (5.4)	14.5 (5.5)	13.4 (5.2)

**Table 3 ijerph-18-12341-t003:** Item loadings on corresponding factors as revealed by exploratory factor analysis of the Internet Addiction Test (IAT) in the school children sample.

Items		Extracted Factors			
		Factor 1	Factor 2	Factor 3	Factor 4
1	Do you find that you stay online longer than you intended?	0.208	**0.612**	0.062	0.192
2	Do you neglect household chores to spend more time online?	0.224	**0.527**	0.150	−0.002
3	Do you prefer excitement of the Internet to intimacy with your partner?	**0.448**	0.284	0.288	0.041
4	Do you form new relationships with fellow online users?	0.234	0.131	**0.527**	−0.091
5	Do others in your life complain to you about the amount of time you spend online?	0.226	0.373	0.143	0.183
6	Does your work suffer (e.g., postponing things, not meeting deadlines, etc.) because of the amount of time you spend online?	0.045	0.099	**0.531**	0.182
7	Do you check your E-mail before something else that you need to do?	0.045	0.168	0.175	0.074
8	Does your job performance or productivity suffer because of the Internet?	0.148	**0.420**	0.233	**0.402**
9	Do you become defensive or secretive when anyone asks you what you do online?	0.270	0.115	0.162	0.254
10	Do you block disturbing thoughts about your life with soothing thoughts of the Internet?	0.174	0.044	**0.481**	0.189
11	Do you find yourself anticipating when you go online again?	**0.572**	0.328	0.078	0.146
12	Do you fear that life without the Internet would be boring, empty and joyless?	**0.523**	0.118	0.243	0.098
13	Do you snap, yell, or act annoyed if someone bothers you while you are online?	**0.403**	0.232	0.113	0.272
14	Do you lose sleep due to late night log-ins?	**0.443**	**0.329**	0.272	0.157
15	Do you feel preoccupied with the Internet when off-line or fantasise about being online?	**0.628**	0.196	0.190	0.148
16	Do you find yourself saying “Just a few more minutes” when online?	**0.578**	**0.532**	0.012	0.180
17	Do you try to cut down the amount of time you spend online and fail?	**0.361**	**0.329**	0.162	**0.381**
18	Do you try to hide how long you’ve been online?	**0.384**	0.195	0.139	**0.451**
19	Do you choose to spend mor e time online over going out with others?	**0.614**	0.189	0.092	0.185
20	Do you feel depressed, moody, or nervous when you are offline, which goes away once you are back online?	**0.538**	0.127	**0.305**	**0.413**

Values in boldface represent loading values above 0.3.

**Table 4 ijerph-18-12341-t004:** Invariance of factor structures of the Internet Addiction Test across groups of eating disorder.

Model	Invariance Levels	χ^2^	Df	*p*	Δχ^2^	Δdf	*p*(Δχ^2^)	CFI	ΔCFI	TLI	ΔTLI	RMSEA	ΔRMSEA	SRMR
Model 2	Configural	759.986	332	0.001				0.897		0.882		0.103		0.0461
	Metric	775.256	351	0.001	15.270	19	0.705	0.898	−0.001	0.890	−0.008	0.100	0.003	0.0481
	Strong	785.181	352	0.001	9.925	1	0.002	0.896	0.002	0.888	0.002	0.101	−0.001	0.0974
	Strict	812.099	376	0.001	26.918	24	0.308	0.895	0.001	0.894	−0.006	0.098	0.003	0.1038
Model 4	Configural	690.297	300	0.001				0.899		0.885		0.104		0.0438
	Metric	707.016	318	0.001	16.719	18	0.453	0.900	−0.001	0.892	−0.007	0.101	0.003	0.0467
	Strong	717.577	319	0.001	10.561	1	0.001	0.897	0.003	0.890	0.002	0.102	−0.001	0.0940
	Strict	740.32	340	0.001	22.743	21	0.358	0.897	0.000	0.896	−0.006	0.099	0.003	0.0992
Model 6	Configural	713.298	328	0.001				0.907		0.893		0.099		0.0456
	Metric	724.393	346	0.001	11.095	18	0.890	0.909	−0.002	0.900	−0.007	0.095	0.004	0.0471
	Strong	737.242	349	0.001	12.849	3	0.005	0.907	0.002	0.898	0.002	0.096	−0.001	0.0974
	Strict	764.561	374	0.001	27.319	25	0.340	0.906	0.001	0.905	−0.007	0.093	0.003	0.1038
Model 8	Configural	587.141	292	0.001				0.924		0.911		0.091		0.0412
	Metric	597.180	309	0.001	10.039	17	0.902	0.926	−0.002	0.918	−0.007	0.088	0.003	0.0430
	Strong	610.667	312	0.001	13.487	3	0.004	0.923	0.003	0.916	0.002	0.089	−0.001	0.0942
	Strict	634.998	336	0.001	24.330	24	0.443	0.923	0.000	0.921	−0.005	0.086	0.003	0.0994
Model 11	Configural	596.787	292	0.001				0.921		0.908		0.093		0.0412
	Metric	610.115	309	0.001	13.329	17	0.714	0.922	−0.001	0.914	−0.006	0.090	0.003	0.0427
	Strong	623.974	312	0.001	13.858	3	0.003	0.919	0.003	0.912	0.002	0.091	−0.001	0.0950
	Strict	648.061	336	0.001	24.087	24	0.457	0.919	0.000	0.918	−0.006	0.088	0.003	0.1000
Model 17	Configural	207.656	98	0.001				0.952		0.936		0.096		0.0433
	Metric	217.050	108	0.001	9.394	10	0.495	0.953	−0.001	0.942	−0.006	0.091	0.005	0.0430
	Strong	232.192	111	0.001	15.142	3	0.002	0.947	0.006	0.938	0.004	0.095	−0.004	0.0892
	Strict	251.243	127	0.001	19.051	16	0.266	0.946	0.001	0.944	−0.006	0.090	0.005	0.0946
Model 19	Configural	20.234	14	0.123				0.994		0.986		0.061		0.0289
	Metric	23.825	19	0.203	3.591	5	0.610	0.995	−0.001	0.992	−0.006	0.046	0.015	0.0319
	Strong	36.294	20	0.014	12.470	1	0.001	0.984	0.011	0.975	0.017	0.082	−0.036	0.0749
	Strict	52.669	28	0.003	16.374	8	0.037	0.975	0.009	0.973	0.002	0.085	−0.003	0.0780

χ^2^: chi-square; df: degrees of freedom; CFI: comparative fit index; TLI: Tucker–Lewis index; RMSEA: root mean square error of approximation; CI: confidence interval; SRMR: standardized root mean residual.

**Table 5 ijerph-18-12341-t005:** Internal consistency, criterion validity, and normality tests of the Internet Addiction Test (20 items, 12 items, and six items) in EDs samples.

Criteria	Whole Sample (*N* = 123)			Anorexia Nervosa (*N* = 59)			Other Eating Disorders (*N* = 64)		
	20 Items	12 Items	Six Items	20 Items	12 Items	Six Items	20 Items	12 Items	Six Items
Coefficient alpha	0.989	0.983	0.972	0.981	0.971	0.950	0.992	0.988	0.980
Range of corrected item-total correlations	0.837–0.937	0.831–0.943	0.883–0.946	0.718–0.898	0.728–0.896	0.816–0.897	0.891–0.957	0.878–0.965	0.904–0.980
Range of Cronbach’s alpha if-item-deleted	All values = 0.989	0.981–0.983	0.963–0.970	0.980–0.982	0.968–0.971	0.936–0.944	All values = 0.992	0.986–0.988	0.973–0.979
Correlation with the Bergen Facebook Addiction Scale	0.906	0.883	0.878	0.856	0.824	0.823	0.908	0.888	0.881
Correlation with the original Internet Addiction Test	--	0.983	0.973	--	0.977	0.972	--	0.984	0.970
Shapiro–Wilk W	0.827	0.811	0.807	0.859	0.830	0.842	0.820	0.815	0.799

For all correlations and Shapiro–Wilk W, *p* < 0.001.

**Table 6 ijerph-18-12341-t006:** Goodness-of-fit indices for models representing the 12-item Internet Addiction Test (12-item IAT) and the six-item IAT in confirmatory factor analysis.

Models	χ^2^	*p*	*df*	CFI	TLI	RMSEA	RMSEA 90% CI	SRMR
Model 16 (C); 2F 12 items	478.247	0.001	53	0.903	0.879	0.085	0.078 to 0.092	0.0560
Model 17 (E); 2F 12 items	329.008	0.001	50	**0.936**	**0.916**	**0.071**	0.063 to 0.078	0.0458
Model 18 (C); 1F six items	160.494	0.001	9	0.899	0.832	0.123	0.106 to 0.140	0.0506
Model 19 (E); 1F six items	35.038	0.001	7	**0.981**	0.960	**0.060**	0.041 to 0.080	0.0241

χ^2^: chi-square; df: degrees of freedom; CFI: comparative fit index; TLI: Tucker–Lewis index; RMSEA: root mean square error of approximation; CI: confidence interval; SRMR: standardized root mean residual; F: factor; (C): crude model; (E): the model involves correlating residuals. Values in bold denote acceptable/good fit.

**Table 7 ijerph-18-12341-t007:** Invariance of 12-item IAT across female groups (women with EDs vs. healthy university student).

Groups	Invariance Levels	χ^2^	df	*P*	Δχ^2^	Δdf	*p*(Δχ^2^)	CFI	ΔCFI	TLI	ΔTLI	RMSEA	ΔRMSEA	SRMR
All EDs	Configural	369.624	100	0.001				0.939		0.920		0.066		0.0202
	Metric	406.460	110	0.001	36.836	10	0.001	0.933	0.006	0.920	0.000	0.066	0.000	0.0259
	Strong	532.650	113	0.001	126.189	3	0.001	0.906	**0.027**	0.890	**0.030**	0.078	−0.012	0.1598
	Strict	978.087	128	0.001	445.438	15	0.001	0.809	**0.097**	0.803	**0.087**	0.104	**0.026**	0.3331
AN	Configural	318.615	100	0.001				0.925		0.901		0.063		0.0406
	Metric	344.099	110	0.001	25.483	10	0.005	0.920	0.005	0.904	−0.003	0.062	0.001	0.0669
	Strong	374.675	113	0.001	30.576	3	0.001	0.911	0.009	0.896	0.008	0.065	−0.003	0.1704
	Strict	604.347	128	0.001	229.672	15	0.001	0.837	**0.074**	0.832	**0.064**	0.082	**−0.023**	0.3300
Other EDs	Configural	324.543	100	0.001				0.936		0.916		0.064		0.0211
	Metric	356.586	110	0.001	32.043	10	0.001	0.930	0.006	0.916	0.000	0.064	0.000	0.0270
	Strong	478.600	113	0.001	122.014	3	0.001	0.896	**0.034**	0.879	**0.037**	0.077	−0.013	0.2125
	Strict	729.810	128	0.001	251.210	15	0.001	0.829	**0.067**	0.824	**0.055**	0.092	**0.015**	0.3945

χ^2^: chi-square; df: degrees of freedom; CFI: comparative fit index; TLI: Tucker–Lewis index; RMSEA: root mean square error of approximation; CI: confidence interval; SRMR: standardized root mean residual; values in boldface signify variance.

**Table 8 ijerph-18-12341-t008:** Invariance of six-item IAT across female groups (women with EDs vs. healthy university student).

Groups	Invariance Levels	χ^2^	df	*P*	Δχ^2^	Δdf	*p*(Δχ^2^)	CFI	ΔCFI	TLI	ΔTLI	RMSEA	ΔRMSEA	SRMR
All EDs	Configural	43.378	14	0.001				0.983		0.963		0.059		0.0170
	Metric	53.515	19	0.001	10.137	5	0.071	0.979	0.004	0.968	−0.005	0.055	0.004	0.0194
	Strong	123.389	20	0.001	69.874	1	0.001	0.938	**0.041**	0.908	**0.040**	0.092	**−0.037**	0.1184
	Strict	406.563	28	0.001	283.174	8	0.001	0.775	**0.163**	0.758	**0.150**	0.149	**0.057**	0.3245
AN	Configural	34.186	14	0.002				0.980		0.957		0.051		0.0316
	Metric	42.937	19	0.001	8.751	5	0.119	0.976	0.004	0.963	−0.006	0.048	0.003	0.0467
	Strong	51.143	20	0.001	8.206	1	0.004	0.969	0.007	0.954	0.009	0.053	−0.005	0.1154
	Strict	212.966	28	0.001	161.823	8	0.001	0.817	**0.152**	0.804	**0.150**	0.110	**0.057**	0.3329
Other EDs	Configural	39.297	14	0.001				0.980		0.957		0.057		0.0136
	Metric	48.712	19	0.001	9.415	5	0.094	0.977	0.003	0.963	−0.006	0.053	0.004	0.0162
	Strong	126.688	20	0.001	77.975	1	0.001	0.916	**0.061**	0.874	**0.089**	0.098	**−0.045**	0.1556
	Strict	279.151	28	0.001	152.464	8	0.001	0.803	**0.113**	0.789	**0.085**	0.127	**0.029**	0.3778

χ^2^: chi-square; df: degrees of freedom; CFI: comparative fit index; TLI: Tucker–Lewis index; RMSEA: root mean square error of approximation; CI: confidence interval; SRMR: standardized root mean residual; values in boldface signify variance.

**Table 9 ijerph-18-12341-t009:** Internal consistency, criterion validity, and normality tests of the Internet Addiction Test (20 items, 12 items, and six items) in a sample of university students (*N* = 1119).

Criteria	20 Item	12 Items	Six Items
Coefficient alpha	0.899	0.865	0.768
Range of corrected item-total correlations	0.207–0.643	0.354–0.664	0.429–0.573
Range of Cronbach’s alpha if-item-deleted	0.891–0.902	0.846–0.860	0.717–0.754
Correlation with time spent online	0.227	0.218	0.206
Correlation with years of internet use experience	0.134 **	0.102 **	0.103 **
Correlation with perceived effect of internet use on academic performance	−0.078 *	−0.107 **	−0.106 **
Correlation with the original Internet Addiction Test	--	0.959	0.923
Shapiro–Wilk W	0.987	0.988	0.986

*: *p* < 0.01, **: *p* < 0.01, for all other correlations and Shapiro–Wilk W, *p* < 0.001.

## Data Availability

The datasets used to produce the current article are publicly available at: https://zenodo.org/record/1286048#.YVYtmppBw2w [73] (access on 27 September 2021)., https://data.mendeley.com/datasets/d5z2bnnv65/1 [47] access on 27 September 2021)., and https://doi.org/10.7910/DVN/QBAFVL [75] (access on 27 September 2021).

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
