# Peer review of "The Six-Item Version of the Internet Addiction Test: Its Development, Psychometric Properties, and Measurement Invariance among Women with Eating Disorders and Healthy School and University Students"

_ijerph, 2021, doi:10.3390/ijerph182312341_

Round 1

Reviewer 1 Report

The Six-item Version of the Internet Addiction 2 Test: Its development, psychometric properties, 3 and measurement invariance among women with 4 eating disorders and healthy school and university students is a valuable research paper that I recommend for publication in this esteemed journal.

The introduction is written clearly and concisely and includes the conceptualization of all key concepts such as IA with a special focus on structure, metric characteristics and invariance as well as eating disorder and even the relations between them.

The research includes research methods on three previously collected data sets in open access, which gives additional importance to the available collected data.

Considering the three separate studies, the presentation of the results is extensive and with a large number of tables and supplement materials. Although extensive material, the results are clearly presented and scientifically relevant.

The discussion is very informative, connects the results of the research with the available findings, but also critically considers some aspects of the scale itself (for example, the use of e-mail lines 630-635).

The conclusion is the most modest part of the paper, it is necessary to state in more detail what new findings are based on research results and what their future contribution is especially in light of the current covid 19 pandemic and the use of the Internet.

The part that is missing considering the current situation is a review of the impact of COVID 19 and IA in this particular population, stated in the keywords.

Some other comments below: 

The abstract is too long and not in line with author guidelines

Recommendation to reduce the number of keywords

2.1.1 detail description of the sample is missing, how many children from 9 schools, mean age? How the research was conducted in the original study?

line 253-258 in this part, which describes the aim of the research, clearly states the purpose of including a sample of university students and women with ED, but it is not clear why the study included the sub-cause of school-age children, which should be stated in this part.

line 290-291 this is a rather low cut of criterion, needs to be explained in more detail and supported by existing research and references

2.2.1. A detailed description of the sample exp. the average age is missing. Since the research also includes girls, are there any additional ethical standards respected in the original research, eg informed parental consent.

2.3.1. A detailed description of the sample, number of participants, average age, etc. is missing.

Characteristic of participants 3.1.1, 3.2.1, 3.3.1. replace from the results to the appropriate section of the research method

Interesting research, which uses already existing data and brings new scientific knowledge is exactly what should be more clearly emphasized in the conclusion part.

Author Response

Manuscript ID: ijerph-1430596

Response to the comments of Reviewer 1

We are very much grateful for the time and sincere help of the Reviewer. We have modified the whole manuscript taking into account the comments of the Reviewer, which are addressed line-by-line as shown below. Replies come underneath in red.

General comments

The part that is missing considering the current situation is a review of the impact of COVID 19 and IA in this particular population, stated in the keywords.

Yes, we have expanded the introduction section to include more details on the effect of the COVID-19 pandemic on IA (line 78-82 and 86-109) as well as emotional eating and EDs (line 243-248).

Some other comments below: 

  1. The abstract is too long and not in line with author guidelines

We agree with the reviewer; the abstract was really too long. We have considerably reduced the length of the abstract. In this version (line 33-55).

  1. Recommendation to reduce the number of keywords

We have reduced the number keywords according to the guidelines.

  1. Characteristic of participants 3.1.1, 3.2.1, 3.3.1. replace from the results to the appropriate section of the research method
  2. 1.1 detail description of the sample is missing, how many children from 9 schools, mean age? How the research was conducted in the original study?
  3. 3.1. A detailed description of the sample, number of participants, average age, etc. is missing.
  4. 2.1. A detailed description of the sample exp. the average age is missing. Since the research also includes girls, are there any additional ethical standards respected in the original research, eg informed parental consent.

Authors’ response: Yes, we have moved “Characteristic of participants 3.1.1, 3.2.1, 3.3.1.” from the results to the appropriate section of the research method. Accordingly, missing details of the sample (e.g., number of participants, average age, etc.) have been provided in the location indicated in comments 4-6. We have also provided details on inquiries of the reviewer such as informed parental consent in 2.2.1. (line 413-414). Thank you so much.

  1. line 253-258 in this part, which describes the aim of the research, clearly states the purpose of including a sample of university students and women with ED, but it is not clear why the study included the sub-cause of school-age children, which should be stated in this part.

Yes, thank you so much for a far-sighted comment. We have modified the text to elaborate more on IA among school children/adolescents (line 134-148) and justified the inclusion of a school-age children in the study (line 308-312)

  1. line 290-291 this is a rather low cut of criterion, needs to be explained in more detail and supported by existing research and references

Yes, we have expanded the criteria and supported by existing research and references (line 376-392).

  1. Interesting research, which uses already existing data and brings new scientific knowledge is exactly what should be more clearly emphasized in the conclusion part.
  2. The conclusion is the most modest part of the paper, it is necessary to state in more detail what new findings are based on research results and what their future contribution is especially in light of the current covid 19 pandemic and the use of the Internet.

We are really grateful for the insightful and kind directions instilled by the reviewer. We have rewritten the conclusion taking into consideration new findings of the research and future contribution is especially in light of the current covid 19 pandemic and the use of the Internet (line 883-902).

We hope that the manuscript has been satisfactorily modified and that the current version will be suitable for publication.

Best regards,

Reviewer 2 Report

The present manuscript deals with a current and relevant theme in the world scenario; the work is well written and can be published in this journal. However, I suggest to the authors some notes that can improve the quality of the manuscript.

At the end of the introduction, I recommend inserting the research hypotheses more clearly because they are not in the text. Don't forget to bring them again to the "discussion" topic.

Regarding the method, although the authors have inserted the inclusion criteria for the participants, you also need to insert the exclusion criteria. It was unclear to me how the participants were recruited, which needs to be presented in the text.

The authors could also describe the data analysis in more detail, as some statistical components are important for the readers to understand these analyses' steps.

Finally, I recommend that the authors insert future directions for further studies in the text's conclusion.

Author Response

Manuscript ID: ijerph-1430596

Response to the comments of Reviewer 2

We are very much grateful for the time and sincere help of the Reviewer. We have modified the whole manuscript taking into account the comments of the Reviewer, which are addressed line-by-line as shown below. Replies come underneath in red.

At the end of the introduction, I recommend inserting the research hypotheses more clearly because they are not in the text. Don't forget to bring them again to the "discussion" topic.

Thank you so much. Yes, we have inserted the research hypotheses more clearly at the end of the Introduction (line 320-334) and mentioned them again at several locations in the discussion.

Regarding the method, although the authors have inserted the inclusion criteria for the participants, you also need to insert the exclusion criteria. It was unclear to me how the participants were recruited, which needs to be presented in the text.

In this version, we described the exclusion criteria more clearly in the description of study 1 (line 365-366) and study 2  (line 414-415). However, the original report pertaining to the university student sample did not provide information on these criteria. We have noted this in the limitation (line 862-864).

The authors could also describe the data analysis in more detail, as some statistical components are important for the readers to understand these analyses' steps.

Yes, we have considerably revised the statistical analysis section, ensuring to clarify vague statistical components.

Finally, I recommend that the authors insert future directions for further studies in the text's conclusion.

Yes, thank you so much. We have inserted future directions for further studies in the text's conclusion (line 883-902).

We hope that the manuscript has been satisfactorily modified and that the current version will be suitable for publication.

Best regards,

Reviewer 3 Report

Thank you for the opportunity to read this quite interesting paper. The material is quite dense which may not be a huge surprise given the nature of the work. This might act as a barrier to the clinical community as opposed to the research community although I think the clinical implications are quite cautious at this stage. You have been wise to point that out in the discussion.

At line 173 /174- IA is high among individuals with comorbidity - some citations to support that might be useful = For example- 

ao, T., Li, M., Hu, Y., Qin, Z., Cao, R., Mei, S., & Meng, X. (2020). When adolescents face both Internet addiction and mood symptoms: A cross-sectional study of comorbidity and its predictors. Psychiatry research, 284, 112795.

Masi, G., Berloffa, S., Muratori, P., Paciello, M., Rossi, M., & Milone, A. (2021). Internet addiction disorder in referred adolescents: a clinical study on comorbidity. Addiction Research & Theory, 29(3), 205-211.

I was pleased to see the material on invariance / variance as I think it is important for the work

Starting at line 211, the material on ED begins. This matters but I am of the view that this discussion is not as vibrant as it needs to be. The case for the linkage between ED and IA needs to be stronger. I recognize the recent literature is not large, but this linkage is so central to your work. Some other material  might include

Tayhan Kartal, F., & Yabancı Ayhan, N. (2021). Relationship between eating disorders and internet and smartphone addiction in college students. Eating and Weight Disorders-Studies on Anorexia, Bulimia and Obesity, 26(6), 1853-1862.

Hinojo-Lucena, F. J., Aznar-Díaz, I., Cáceres-Reche, M. P., Trujillo-Torres, J. M., & Romero-Rodríguez, J. M. (2019). Problematic internet use as a predictor of eating disorders in students: a systematic review and meta-analysis study. Nutrients, 11(9), 2151.

In study 1, given your comments above regarding cross cultural validation are you satisfied this is standardized enough for this population?

In table 3 - Academic grade First grade Second grade Third grade Fourth grade  - is this referring to years in university - if so this language is more about elementary school. I was a bit confused here.

Overall this is a good paper. The use of 3 somewhat unrelated studies makes consumption of the paper more challenging. I might suggest that, at the beginning of the methods section you add some material more closely linking the 3 studies together. 

Author Response

Manuscript ID: ijerph-1430596

Response to the comments of Reviewer 3

We are very much grateful for the time and sincere help of the Reviewer. We have modified the whole manuscript taking into account the comments of the Reviewer, which are addressed line-by-line as shown below. Replies come underneath in red.

At line 173 /174- IA is high among individuals with comorbidity - some citations to support that might be useful = For example- 

ao, T., Li, M., Hu, Y., Qin, Z., Cao, R., Mei, S., & Meng, X. (2020). When adolescents face both Internet addiction and mood symptoms: A cross-sectional study of comorbidity and its predictors. Psychiatry research284, 112795.

 Masi, G., Berloffa, S., Muratori, P., Paciello, M., Rossi, M., & Milone, A. (2021). Internet addiction disorder in referred adolescents: a clinical study on comorbidity. Addiction Research & Theory29(3), 205-211.

Yes, we have expanded the text on IA and comorbidity, especially among adolescents (line 134-148, and 196). Thank you so much for the rich resources. 

Starting at line 211, the material on ED begins. This matters but I am of the view that this discussion is not as vibrant as it needs to be. The case for the linkage between ED and IA needs to be stronger. I recognize the recent literature is not large, but this linkage is so central to your work. Some other material  might include

Tayhan Kartal, F., & Yabancı Ayhan, N. (2021). Relationship between eating disorders and internet and smartphone addiction in college students. Eating and Weight Disorders-Studies on Anorexia, Bulimia and Obesity26(6), 1853-1862.

Hinojo-Lucena, F. J., Aznar-Díaz, I., Cáceres-Reche, M. P., Trujillo-Torres, J. M., & Romero-Rodríguez, J. M. (2019). Problematic internet use as a predictor of eating disorders in students: a systematic review and meta-analysis study. Nutrients11(9), 2151.

Yes, thank you so much for such helpful resources. Accordingly, we have expanded the text addressing the linkage between ED and IA, especially among students (line 243-248, 255-273, and 282-288).

In study 1, given your comments above regarding cross cultural validation are you satisfied this is standardized enough for this population?

The reviewer is quite insightful raising such an important issue. We believe that cross cultural validation of the scale is not standardized enough for this population. We have noted this in the limitation (line 857-861).

In table 3 - Academic grade First grade Second grade Third grade Fourth grade  - is this referring to years in university - if so this language is more about elementary school. I was a bit confused here.

 Thank you so much. Yes, we have used proper terms to describe academic grades in the university student sample in Table 3.

Overall this is a good paper. The use of 3 somewhat unrelated studies makes consumption of the paper more challenging. I might suggest that, at the beginning of the methods section you add some material more closely linking the 3 studies together. 

Yes, thank you so much. Accordingly, we have a brief text at the beginning of the methods section that attempted to more closely link the 3 studies together (line 339-356). 

We hope that the manuscript has been satisfactorily modified and that the current version will be suitable for publication.

Best regards,

Round 2

Reviewer 2 Report

All the requested changes were made by the authors.

Author Response

Dear Reviewer 2,

Thank you so much for your helpful comments, which really made a difference in the quality of the current version of the manuscript.

Best regards,
